# Novel Methylation Biomarkers for Colorectal Cancer Prognosis

**DOI:** 10.3390/biom11111722

**Published:** 2021-11-19

**Authors:** Alvaro Gutierrez, Hannah Demond, Priscilla Brebi, Carmen Gloria Ili

**Affiliations:** Millennium Institute on Immunology and Immunotherapy, Laboratory of Integrative Biology (LIBi), Centro de Excelencia en Medicina Traslacional (CEMT), Scientific and Technological Bioresource Nucleus (BIOREN), Universidad de La Frontera, Temuco 4810296, Chile; a.gutierrez05@ufromail.com (A.G.); hannah.demond@ufrontera.cl (H.D.)

**Keywords:** colorectal cancer, methylation, prognosis, biomarkers

## Abstract

Colorectal cancer (CRC) comprises the third most common cancer worldwide and the second regarding number of deaths. In order to make a correct and early diagnosis to predict metastasis formation, biomarkers are an important tool. Although there are multiple signaling pathways associated with cancer progression, the most recognized are the MAPK pathway, p53 pathway, and TGF-β pathway. These pathways regulate many important functions in the cell, such as cell cycle regulation, proliferation, differentiation, and metastasis formation, among others. Changes in expression in genes belonging to these pathways are drivers of carcinogenesis. Often these expression changes are caused by mutations; however, epigenetic changes, such as DNA methylation, are increasingly acknowledged to play a role in the deregulation of oncogenic genes. This makes DNA methylation changes an interesting biomarkers in cancer. Among the newly identified biomarkers for CRC metastasis *INHBB*, *SMOC2*, *BDNF*, and *TBRG4* are included, all of which are highly deregulated by methylation and closely associated with metastasis. The identification of such biomarkers in metastasis of CRC may allow a better treatment and early identification of cancer formation in order to perform better diagnostics and improve the life expectancy.

## 1. Introduction

Colorectal cancer (CRC) is the third most common cancer worldwide and the second most common regarding global cancer-related deaths [1]. Most CRC cases occur sporadically, with risk factors including environmental and food-borne mutagens as well as chronic intestinal inflammation [2,3,4]. Hereditary factors are only found in a minority of cases but are nevertheless important to consider [5,6,7]. Most cases of CRC are thought to occur through a stepwise accumulation of mutations, encompassing the repression of tumor suppressor genes and activation of oncogenes [8]. At the time of diagnosis, 20% of the patients, with CRC, are presenting metastasis. In addition, 35–45% of the patients with localized CRC show recurrence within 5 years [9]. Metastasis usually results in conventional therapies becoming inadequate, and in consequence is associated with a poor 5-years survival rate of <10% [10,11]. It is still unclear what exactly causes a carcinoma to metastasize, although recent studies have associated key genes such as *APC*, *K-RAS,* and *SMAD4*, among others in CRC metastasis formation [12]. Understanding the metastasis process will allow the identification of molecular biomarkers, which are important tools for early diagnosis and prognosis of CRC, as early prognosis is currently the most successful and cost-effective way to enhance survival rate of CRC patients.

In the last decades, cancer diagnosis and prognosis at early stages have improved resulting in an increase of cure rates. New screening tools have been developed, focusing on early detection of cancer [8,9]. Biomarkers are used to distinguish between normal or abnormal (cancer) status in patients and to provide relevant information about disease diagnosis, prognosis, prediction, stratification, and therapy monitoring and improving treatment in cancer patients [10]. Usually, biomarkers are referring to DNA mutations, RNA or microRNA expression changes, epigenetic changes, and changes of metabolite abundance [13,14,15]. Their usages depend on multiple factors, including patients, stages of disease, and therapy/treatment. One of the largest challenges in medical diagnosis at the moment is to understand the molecular mechanisms of metastasis formation. This will aid the discovery of new and better ways to predict and diagnostic early metastasis events. One new and interesting method uses tumor budding as a predictive biomarker for distant metastasis. However, the scoring system needs to be optimized for it to be reliably used and it requires invasive colonoscopy [16]. Many studies focus on biomarkers that can be identified in blood plasma or stool samples, as these can be obtained in a less invasive way, while still having high sensitivity and accuracy to diagnose CRC and metastasis While several biomarkers for CRC diagnosis are available, it is not that easy to diagnose or predict metastasis. Identifying epigenetic changes, such as DNA methylation changes that are specific for metastasis, may help the diagnosis and prognosis of CRC [17].

The use of biomarkers has become robust and efficient method to evaluate potential abnormal biological processes such as cancer [14]. Using DNA methylation changes as biomarkers depends on the sensitivity of the detection method to distinguish between unmethylated cytosines from methylated cytosines. A tumor sheds cells, which are degraded, and DNA fragments from these cells can be detected in the blood or stool of patients. However, as these fragments only form a tiny fraction of the total DNA found in blood and stool, both the biomarker and method need to be carefully selected. The biomarker needs to show a very high methylation change (for example 0% methylation in normal tissue and 100% methylation in tumor tissue). The change must also be very homogeneous across tumor cells and patients. The method is preferably quantitative, such as qPCR, rather than qualitative like regular PCR, to detect methylation changes, as only a small fraction of DNA will show the change. One of the most common ways to detect DNA methylation changes is using bisulfite conversion of DNA. Treating DNA with bisulfite leads to the deamination of cytosines to uracil, while methylated cytosines remain unchanged [18]. The methods to detect DNA methylation in cancer are constantly evolving, trying to solve limitations associated with DNA quantification and repeatability [19].

In this review, we describe our current understanding of the mechanisms leading to colorectal cancer and metastasis formation and discuss the role of DNA methylation in these pathways. We show how our increasing knowledge in these processes has led to the identification of biomarkers for the prognosis of CRC and metastasis. We present and discuss novel biomarkers that have the potential to change our treatment strategies for CRC in the future.

## 2. Search Strategy

The literature search was performed with PubMed using the following search terms: “Cancer”, “Colorectal Cancer”, “Biological Markers”, “Biomarkers”, “DNA methylation”, and “Methylation marker”. Animal studies were excluded, as well articles and reviews written in languages other than English. The search yielded a total of 1245 papers, of which 306 were found relevant. These 306 articles were further selected after full-text reading, leaving 297 articles. Based on our inclusion and exclusion criteria, another 209 articles were eliminated, resulting in 88 articles selected for this review (Figure 1). The novel biomarkers discussed in Section 6 were selected based on the following criteria: (1) multiple studies have highlighted their deregulation in CRC and other cancers, (2) studies have associated their deregulation to methylation changes, and (3) their deregulation has been linked to poor prognosis and metastasis formation. This list is a selection of interesting candidates and by no means exhaustive.

## 3. Insights into Colorectal Carcinogenesis

In the majority of CRC cases (83%) [20], the process of adenoma formation is initiated by a loss-of-function mutation genes such as tumor suppressor gene *TP53*, oncogenes *KRAS* and *PIK3CA* [13], or *APC.* The loss of *APC* function leads to activation of the WNT signaling pathway [21]. The canonical Wnt pathway is normally regulated through to the phosphorylation of β-catenin by the “destruction complex” composed of *APC*, glycogen synthase kinase 3β (GSK3β), Axin, and casein kinase 1 (CK1). Phosphorylation marks β-catenin for subsequent ubiquitination and degradation in the proteosome. In absence of functional *APC*, β-catenin is not degraded, allowing for the translocation of β-catenin to the nucleus, where it activates tumorigenic genes [22,23].

CRC formation is usually driven by the activation of three well-characterized pathways: the MAPK pathway, p53 pathway, and TGF-β pathway [9]. The MAPK pathway plays a crucial role in the regulation of cell proliferation, differentiation, and stress response [24]. There are several MAPK protein family members, which can trigger different disorders [25]. The p53 pathway acts in response to cellular stress, targeting genes involved in apoptosis, cell-cycle arrest, DNA repair, and senescence [18,19,20,21]. The p53 pathway can be activated by at least three different mechanisms [26,27]: DNA damage [28,29], oncogene expression such RAS or Myc [30], and chemotherapeutic drugs or protein-kinase inhibitors [29,31]. TGF-β corresponds to a superfamily of cytokines composed of 33 members [32], comprising TGF-β1, -β2, and -β3, as well as activins, inhibins, growth differentiation factors (GDFs), and bone morphogenic proteins (BMPs) [33,34]. Aberrant activation of TGF-β results in cell-cycle modifications [35] and is involved in immune suppression, growth inhibition, epithelial-to-mesenchymal transition (EMT), and cell migration, among others [34]. These effects are caused by the three isoforms of TGF-β through two possible mechanisms, Smad and non-Smad signaling [36]. TGF-β ligands bind to their corresponding receptors, which marks to beginning of a signaling cascade, which is initiated by type II receptors (TβRII), functioning as an activator of the type I receptor (TβRI), by phosphorylating its glycine/serine-rich GS domain. When activated, TβRI transduces the signal downstream [32,36] through Smad signaling. Smad proteins are classified into three groups: R-Smads, Co-Smad, and I-Smad. Receptor-activated Smads (R-Smads) are comprised of Smad-1-5-8 and Smad-2-3 and are phosphorylated and activated by different receptors, including TβR, ActR, and BMPR, [37]. The second group has the common mediator Smad (Co-Smad or Smad4), which binds and forms a complex with R-Smads and they translocate to the nucleus, where they bind to other transcriptional receptors to enhance their DNA affinity. The third group are Smad inhibitors (I-Smad) and include Smad-6, which inhibits BMP signaling, and Smad-7, which can inhibit both BMP and TGF-β [38]. In CRC, members of the TGF-β/Smad-4 complex are among the most important biomarkers listed in Table 1. They are involved in DNA binding and gene expression regulation [39], and include genes such as *SNAIL*, *SLUG*, *TWIST,* and *ZEB*, promoting EMT formation and genes important for DNA damage response, (e.g., *ATM*, *MSH2*, and *BRCA1*) [32,33].

## 4. From CRC to Metastasis: A Complex Journey

Metastasis is the final and most devastating stage of cancer. It is the leading cause for fatalities in cancer [75,76]. Furthermore, we still do not fully understand what causes a carcinoma to metastasize. It is, therefore, not surprising that metastasis formation is a highly interesting topic in both research and clinical studies [90]. It appears that metastasis formation is a diverse process, as researchers have found that both the affected organs and the time course differ between cancers. For example, prostate cancer metastasis and ocular melanoma metastasis are restricted mostly to bone and liver, respectively [91]. In contrast, some types of cancer disseminate readily like breast, lungs, and testicles, others progress faster when metastatic cells come from luminal cells and are accompanied with poor survival [92].

Acquisition of a mesenchymal phenotype is an important step during metastasis progression. Pathways such as WNT, TGF-β, or p53 collaborate to drive CRC progression as described before, but the location (organ) where metastasis implants needs to be taken into consideration in the metastasis formation [9]. To explain the mechanism driving a metastatic event, Stephen Paget proposed an interconnection between metastatic cells released and organ or tissue receptors providing the right environment for the cells to invade [93,94]. This theory suggests a collaboration between the released cells and receptor cells and is termed “Seed and Soil” [95]. It implies that every metastatic event occurs only when both “seed” and “soil” groups of cells are compatible to interact together. This theory was later adapted by James Ewing, suggesting that metastasis is a result of purely mechanical factors of cells circulating in the vascular network. However, this theory did not explain the behavior or dissemination in which some organs with small blood supply become hosts of metastatic events and others with considerably more blood supply do not [94]. Nowadays it is thought that it is most likely a combination of both theories that explains the behavior of metastatic cells [96].

One of the key processes in tumor metastasis is thought to be the EMT [95]. This process comprises the early differentiation between epithelial to mesenchymal cells and has multiple consequences, including the modification of normal cell-to-cell junction and apical–basal polarity to become more invasive and loss of apical–basal polarity [97]. EMT is initiated by deregulation of three principal groups of regulators that are involved in the inactivation of E-cadherin, consisting of the bHLH family of transcription factors, the zinc-finger E-box-binding homeobox family, and the Snail zinc-finger family of transcription factors [98]. Together they enhance the effects of N-cadherins, integrin-αvβ6, and upregulate genes involved in the invasion and migratory behavior [94].

WNT signaling is another essential component of metastasis. It is involved in the process through its β-catenin-dependent (canonical) and β-catenin-independent (non-canonical) pathways. In the canonical pathway, WNT is degraded in the cytoplasm and migrates to the nucleus. There it interacts with TCF/LEF transcription factors, leading to abnormal secretion of β-catenin/TCF-responsive target genes, which in turn provoke cell differentiation and apoptosis [99,100]. The β-catenin-independent pathway involves the activation of GTPases, which regulate the activity of pathways like MAPK through the release of intracellular calcium. This results in changes in cell adhesion, cell polarity, and cellular migration [101].

Another important factor in metastasis formation is the TGF-β pathway, which can work through the SMAD-dependent and non-SMAD-dependent pathways. In the SMAD dependent-pathway, the E-cadherin transcriptional repressor induces the modification of SMAD levels [102] through phosphorylation of the C-terminal region of SMAD [34]. Once the SMAD complex is activated, it is translocated to the nucleus where it binds to SMAD-binding DNA elements (SBE), which activate or block the transcription of target genes [103]. In the SMAD-independent pathway, TGF- β activates other mediators such as MAPK, extracellular signal-regulated kinase (Erk), and c-Jun amino-terminal kinase (JNK) [104], which have been implicated in invasive behavior [105], and all of which are known biomarkers related to cell differentiation and promoting cell release.

The EMT phenotype is not only important for the formation of metastasis, but also plays a critical role in the formation and progression of cancer in general. EMT is responsible for the formation of metastasis-initiating cells (MICs), which form a small group of circulating tumors cells (CTCs) [106]. CTCs are tumor cells that enter the bloodstream and are considered to be metastasis precursors displaying EMT-like behavior [107]. CTC development is an intricate process, in which all of the common deregulated mechanisms, such as the PI3K/AKT, MAPK, and ERK pathways [108], as well as macrophage participation in EMT induction [109], are involved. However, CTCs and its capability to migrate is not the final step of invasion. After circulating through the bloodstream and finding a favorable microenvironment, CTCs must colonize a host tissue through a reverse phenotypical event known as mesenchymal-to-epithelial transition (MET) [110]. MET describes the transition from mesenchymal to epithelial cells and provides CTCs with a growth advantage in the new organ [109].

So far, we have described how genomic mutations leading to the disruption of protein–protein interactions, drive carcinogenesis. However, it is important to understand that there are other factors besides point mutations and protein signaling involved in cancer progression. Changes in chromatin structure and epigenetic modification are often observed in cancer [111]. These changes include a multiple of phenotypes that are commonly used to phenotype cancers, such as chromosome instability (CIN), microsatellite instability (MSI), and the CpG island methylator phenotype (CIMP). These phenotypes can be used for prognosis and direct treatment strategies. Furthermore, hypermethylation and hypomethylation profiles have been shown to alter gene expression, therefore contributing to cancer proliferation from CTCs clusters and indicating that epigenetic changes may act as drivers of carcinogenesis and prognostic markers.

## 5. Chromosome Stability and DNA Methylation Profiles: Can Epigenetic Changes Drive Metastasis?

With its relevance in development and disease, DNA methylation of the human genome has become a widely studied object. The human genome consists of over 28 million CpG sites, which are on average 70% methylated [112]. DNA methylation is characterized as the covalent addition of a methyl group (-CH3) at the carbon-5 of cytosine, which results in 5-methylcytosine (5mC) [113,114]. DNA methylation is catalyzed by three DNA methyltransferases (DNMTs), consisting of DNMT1, DNMT3a, and DNMT3b [71,115]. DNMT3a and DNMT3b are de novo DNA methyltransferases and are important during development and cell differentiation [116]. DNMT1 is required for the maintenance of DNA methylation in each cell cycle and methylates hemimethylated DNA after each cell division to ensure that the newly synthesized chromosomes inherit the same methylation pattern as the original ones.

There are three major epigenetic phenotypes in colorectal cancer that can be used to categorize tumors into different subtypes with different prognosis. These include CIN, MSI, and CIMP [117]. The CIN pathway is characterized by chromosomal abnormalities [118], such as aneuploidies, deletions, insertions, and amplifications, and is associated with poor prognosis, metastasis, and therapeutic resistance [21]. It is thought to be caused by changes in chromosome segregation, telomere dysfunction, and DNA damage response, which alter the expression of genes essential for maintenance of normal cell functions, including *K-RAS*, *APC*, *PI3K,* and *TP53*, among others [119]. The MSI pathway is named after the characteristic changes (insertions or deletions) of short repetitive DNA nucleotide sequences, knows as microsatellites, which are important for heterochromatin stability [120,121]. Loss of a functional DNA mismatch repair system (MMR), through mutations of *MSH2*, *MSH3,* or *MSH6*, results in a hypermutable phenotype [122], with MSI tumors often harboring > 1000 mutations. CIMP was first identified by Toyota et al. [123] and described as hypermethylation of CpG surrounding the promoter regions of certain genes in some CRC patients [124,125]. Since then, many more aberrantly methylated CpG islands (CGI) have been found in CRC and added to the CIMP [103]. The phenotype is used as a prognostic marker, with CIMP-positive patients having a poorer prognosis than CIMP-negative patients [103,110]. Among the CIMP genes, there are some interesting candidates, in which aberrant methylation may have functional consequences regarding CRC development and progression [126]. These include the DNA repair enzymes MLH1 and MGMT and the tumor suppressor genes *CDKN2A* and *THBS1* [127].

Besides these three classic phenotypes, the cancer genome commonly undergoes widespread DNA methylation changes. One commonly observed characteristic is global hypomethylation of CpG poor regions, including intergenic regions and gene bodies, which are normally hypermethylated. In contrast, CpG-rich regions, also known as CpG islands (CGIs), are usually hypomethylated. In the process of cancer progression many of these CGIs become hypermethylated. As the majority of gene promoters contain CGIs, this hypermethylation may directly affect gene expression or enforce a repressive state that is caused by other chromatin modifications or changes in transcription factor binding [46]. Indeed, DNA methylation changes often correlate with suppression or inactivation of tumor suppressor genes [128]. These DNA methylation changes are therefore interesting biomarker candidates. For example, promotor methylation at the *TERT* hypermethylated oncological region (THOR) correlates with the upregulation the *TERT* gene affecting telomere maintenance [129]. THOR has therefore been suggested as a prognostic biomarker. To increase sensitivity and specificity, gene panels have been shown to improve detection of methylation changes in blood or stool [130,131,132]. For example, analysis of promoter hypermethylation in *ALX4*, *BMP3*, *NPTX2*, *RARB*, *SDC2*, *SEPT9*, and *VIM* improves CRC detection in blood plasma [46]. Even so, the sensitivity of biomarker assays in blood/stool is often low, which currently still hinders their use in medical treatment.

It is important to keep in mind that CRC is a heterogeneous disease and different patients display different phenotypes associated with different prognosis and treatment options. To summarize all these changes, the international CRC Subtype Consortium was formed, which combined classifications using transcriptomics with models based on molecular and clinical features to generate four consensus molecular subtypes of CRC [133]. Together, these subtypes can categorize 87% of CRC cases. The remaining 13% display mixed features of several subtypes and possibly represent transition phenotypes or intratumor heterogeneity. The categorization of these subtypes depends heavily on epigenetic pathways, demonstrating the importance of epigenetic changes for cancer research CMS1 (MSI Immune), which is characterized by CIMP-high, MSI, BRAF mutation, and gene hypermutation. CMS2 (Canonical) cancers show high CIN and upregulation of the WNT pathway. CMS3 (Metabolic) tumors are enriched for *KRAS* mutations and frequently exhibit MSI, as well as low CIN and low CIMP. CMS4 (Mesenchymal) is characterized by high CIN profiles, activation of TGF-β, and upregulation of EMT. Most CMS4 tumors harbor *KRAS* and *BRAF* mutations [117,133]. Epigenetic profiles are important for the classification of CRC. As most of the aberrant hypermethylation occurs in CpG islands, many of which are overlapping promoter regions enhancers, it was hypothesized that DNA methylation changes may directly affect gene transcription and therefore have functional consequences in carcinogenesis and metastasis formation [134]. However, it is not clear how DNA methylation may inhibit gene expression [135]. One possibility is that DNA methylation may prevent transcription factor binding [136]. DNA methylation may also contribute to heterochromatin formation by recruiting histone modifiers and chromatin remodelers [135]. Independent of their function, DNA methylation changes serve as extremely stable and easily detectable markers of changes in transcriptional state. Furthermore, changes in DNA methylation occur very early in carcinogenesis, making them interesting candidates for the development of biomarkers.

One of the biggest challenges in using CRC DNA methylation biomarkers for non-invasive screening assays is their reliable detection in blood and stool samples. These types of assays require detection of methylation changes in cell-free tumor DNA among the unchanged cellular DNA of the sample. There is currently no general consensus for an adequate technology, which has been suggested to slow implementation of DNA methylation changes in clinical diagnosis [137]. Delvenne et al. assessed different methods for DNA methylation analysis, including bisulphite sequencing, MS-DBA, MS-HRM/SMART-MSP, COBRA, MS-SnuPE, Q-MSP/MethylQuant, MethyLight, mass spectrometry, pyrosequencing, and Illumina Golden Gate [138]. All of them have two major criteria in common: (i) a biological/chemical module by which can be distinguish methylated from unmethylated fragments; and (ii) an amplification step that allows for the detection of methylation changes using a variety of techniques such as DNA sequencing, mass spectroscopy, microarray, and PCR [139]. Even though these techniques are all well established, they have advantages and disadvantages when it comes to DNA methylation biomarker screening. Delvenne et al. identifies bisulfite sequencing as the gold standard regarding sensitivity, but also highlights disadvantages, such as it being a labor-intensive, high-cost method, among others [138]. The lack of appropriate methods is currently hindering the implementation of DNA methylation biomarkers in clinical diagnosis. Method development is therefore one of the key issues that should be addressed.

## 6. Novel Methylation Biomarkers in CRC and Metastasis Prognosis

Biomarkers are important tools for disease detection and prognosis. They are cost-effective and often less invasive than other screens, like for example in the case of CRC colonoscopy. Currently, the majority of biomarkers used in clinical assays are designed for the early detection of cancer. Among the established CRC biomarkers, there are several based on DNA methylation changes, including *NDRG4*, *BMP3* [140], *SEPT9* [141], *BCAT1,* and *IKZF1* [142]. The variety of biomarkers already on the market demonstrates the successful translation from research studies to clinical assays. However, none of these assays cover metastasis prognosis. Recently, novel biomarkers have been described that are aimed at detection of metastasis formation.

It is well known that CGI methylation is affected in CRC, and epigenetic alterations in the pathogenesis of CRC and liver metastasis have been widely reported. When comparing DNA methylation of distinct genes in CRC tumors of different stages (I–IV), including liver metastasis, Ju et al. found distinct DNA methylation in CRC with metastasis [143]. Importantly, in stage IV, methylation profile appeared to be established before metastasis occurred, which would make it an excellent biomarker for early diagnosis of metastasis formation. Specifically, this study found progressively increased DNA methylation levels of *MGMT* and *TIMP3* during the metastatic process. *MGMT* is a DNA repair gene that is frequently methylated in CRC and is correlated with G to A transition mutations in cancer-related genes, such as *KRAS*, *TP53*, and *PIK3CA* [143]. *MGMT* hypermethylation has also commonly been found in brain metastasis from CRC [144]. A study examining genome-wide DNA methylation in CRC tissue of stages I–III and in paired primary and metastatic tumors also found that methylation frequencies of *MGMT* and *TIMP3* were progressively increased during the metastatic process in CRC. Furthermore, *UPK3A* was found to be methylated only in hepatic metastasis [143]. Other genes that displayed a higher incidence of methylation in liver metastases compared to local CRC are *ID4* and *HPP1/TPEF*. The inhibitor of DNA binding 4 (*ID4*), a dominant negative helix-loop-helix protein, acts as a tumor suppressor gene. *ID4* methylation is correlated with histopathological tumor grade and poor prognosis [145]. Methylation of the *HPP1/TPEF* gene was also observed in the early stage of primary CRC (77%) as well as liver metastasis (79%) [146]. Interferon regulatory factor 8 (*IRF8*), a transcription factor of the IRF family, acts as a tumor suppressor gene and is silenced by epigenetic mechanisms in several human tumors [147,148]. In liver metastasis from CRC, *IRF8* was found hypermethylated in 83% of the cases, whereas it was detected in 43% of primary colon carcinomas [149]. Hypermethylation of other genes were associated with lymph node metastasis, such as *CDKN2A/p16* and *CDKN2A/p1445*, *DFNA546,47*, *HLTF23,* and *ESR1* [150].

Recently, genome-wide sequencing studies have helped to study DNA methylation changes in metastasis on a wider scale. A study of 10 paired samples of CRC primary tumors and metastasis, using single-cell multi-omics sequencing analysis, characterized colon cancer cell populations according to their somatic copy number alterations (SCNAs), using both transcription and methylation profiles. The samples showed low methylation levels in CGIs regions of normal tissues, high methylation levels in primary tumors, and intermediate methylation levels in metastasis [151]. Another study in 59 CRC patients (stages I/II and IV), where primary tumor, adjacent normal tissue, and metastatic tumor tissues were analyzed using targeted bisulfite sequencing, identified 23 differentially methylated regions, which were associated with a high probability of liver metastasis of CRC [152]. We recently compared five paired samples from CRC patients (normal adjacent tissue, primary tumor, and lymph node metastasis) using bisulphite sequencing and identified five hypermethylated genes in all tumor samples that were associated with poor prognosis: *BDNF*, *FIGN*, *HCN4*, *HTRA3*, and *STAC2* [88]. In addition, *TBRG4* was found to be hypomethylated exclusively in CRC metastasis samples [88].

### 6.1. Microbiome and DNA Methylation as Biomarkers of Colorectal Cancer

Even though both the gut microbiome and DNA methylation are recognized to have important roles during colon carcinogenesis, only a few studies have looked at the combined function of them in CRC development. However, in the last decade it has emerged that DNA methylation changes correlate with changes in the microbiome, indicating an intricate relationship between the two.

DNA methylation changes of genes involved in early colon carcinogenesis have been associated with *Peptostreptococcus* and *Schwartzia*, and the absence of the *Flavinofractor* genus. Furthermore, the *Firmicutes* phyla has been correlated with *CDH13* methylation [153]. In addition, the commensal bacteria *Bifidobacterium breve* and *Lactobacillus rhamnosus* GG can increase global DNA methylation and decrease histone acetylation in inflammatory bowel disease, one of the risk factors for CRC; therefore, both bacteria are beneficial for the colon [154,155]. A recent study evaluated the effects of butyrate, the fermentation end product of gut microbiota and an anti-tumor reagent, on α-ketoglutarato-mediated epigenetic modification, in colorectal adenocarcinoma HT-29 and Caco-2 cells [156]. There is evidence that butyrate has a direct effect on DNA methylation by regulating key enzymes, including TET and DNMT1 [157]. In HT-29 and Caco-2 cells, it was found that butyrate suppressed proliferation, potentiated differentiation, and induced apoptosis. Moreover, butyrate upregulated acetyl-CoA and α-ketoglutarato, which was concomitant with enhanced histone acetylation and DNA demethylation in the promoter of the DNA mismatch repair (MMR) genes [156]. Butyrate has also been shown to modify DNA methylation and inhibit WNT signaling in human gastric cancer cells, a pathway also known to be activated in CRC [157].

A recent multi-omics study analyzed the microbiome, metabolome, transcriptome, and DNA methylome of paired CRC and normal adjacent tissues [157]. The investigation identified 16 genes to which promoter CGI hypermethylation was related with a downregulation of gene expression. Furthermore, an abundance of *Fusobacterium* was significantly associated with the downregulated expression of *PI16*, *FCRLA,* and *LSP1* in cancer tissues compared to healthy tissues [157]. Interestingly, a relationship between *Fusobacterium* status and epigenetic features of CRC, such as CIMP status and MSI, was previously reported, where *Fusobacterium* enrichment was associated with specific molecular subsets of CRCs, offering support for a pathogenic role in CRC for this bacterium [158]. *Fusobacterium* is able to affect gene expression and/or DNA methylation directly through nucleomodulins. These are bacteria that secrete virulence proteins, which allow them to enter the nucleus of the surrounding colon cells. In tissue samples, *F. nucleatum* intracellular infection has been associated with *CDKN2A* hypermethylation [159]. There are other bacteria using the same pathway [160]. For example, the nuclear traffic of *Acinetobacter baumannii* Tnp induces CGI hypermethylation of *E-cadherin* in COS-17 cells [161]. It remains unclear though how microorganisms, such as *F. nucleatum,* can induce epigenetic changes mechanistically.

### 6.2. Inhibin Subunit Beta B (INHBB)

Inhibin subunit beta B gene (*INHBB*) encodes a pre-protein, which subsequently is processed to inhibin and activin [86]. Inhibin protein consists of a heterodimer formed by the two subunits α and β, which are secreted by ovarian granulosa cells and testicular Sertoli cells [162]. It belongs to TGF-β superfamily, where it is unique, being the only heterodimer in the TGF- β family. Even though the inhibin signaling pathway is not fully known, it acts through antagonizing activin (another TGF- β protein encoded by *INHBB*) by binding to the activin receptor ActRII [163]. This prevents a signaling cascade that results in the downregulation of intracellular phosphorylation, and finally the downregulation of SMAD signaling [164]. In many cancers, including CRC, as well as oral, endometrial, prostate, renal, and thyroid cancer, *INHBB* is overexpressed [163]. Interestingly, *INHBB* is also associated to DNA methylation changes [165,166]. In fact, *INHBB* is hypermethylated in CRC, and its expression was negatively correlated with methylation. Furthermore, the overexpression of *INHBB* is significantly and positively associated with invasion depth, distant metastasis, and CRC stage [86,167].

### 6.3. SPARC-Related Modular Calcium-Binding Protein 2 (SMOC2)

SPARC-related modular calcium-binding protein 2 (SMOC2) is a member of the secreted protein acidic and rich in the cysteine SPARC family of matricellular proteins [168]. SPARC family proteins have been shown to be associated with several biological processes, such as cell adhesion, migration, and tissue repair, among others [169,170]. Moreover, SPARC proteins have been related to the promotion of various tumors, such as glioma, prostate, and gastric carcinomas [171]. Initial studies demonstrated the capability of SMOC2 to promote cell cycle progression and DNA synthesis, as well as impact cell growth in hepatocellular carcinoma [172]. Even though the mechanism of how SMOC2 promotes cell deregulation is not fully understood, recent research has shown that SMOC2 interacts with Fzd6 and LRP6 to activate the Wnt/β-catenin pathway in endometrial CSCs [173]. In CRC, SMOC2 has been suggested as a prognostic marker, having tumor suppressor activity in cancer progression [89]. Finally, the hypermethylation of CpG sites in the promoter region of *SMOC2* suggest an epigenetic regulation in papillary thyroid carcinomas (PTCs) [174]. Whether *SMOC2* is also a good candidate for CRC metastasis prognosis needs to be determined.

### 6.4. Helicase-like Transcription Factor (HLTF)

Helicase-like transcription factor (*HLTF*) is a member of the Switch/sucrose non-fermenting family (SWI/SNF) [175] and is comprised of a DNA-binding domain, a RING helicase motif, and a HIRAN domain [176,177]. *HLTF* has an important function in nucleosome remodeling. It is a mediator in replication fork reversal through its HIRAN domain and can act as a putative tumor suppressor through its role in DNA damage tolerance [178], genome stability maintenance, and gene transcription [58]. It has been suggested that the loss of expression of HLTF in several cancers (colon, gastric, esophageal, cervix, lung, hepatocellular, and bladder) may be caused by promoter methylation, as the downregulation of HLTF is correlated with hypermethylation of the promoter region [86,167,179]. This association makes *HLTF* a candidate as a cancer methylation biomarker [180]. Importantly, *HLTF* and more specifically, *HLTF* methylation, has been related to metastasis formation. Aberrant *HLTF* methylation is thought to affects HTLF’s capability to suppress migration and invasion through the TFG-β/SMAD pathway [58]. Therefore, *HLTF* may be a new methylation biomarker for CRC metastasis prognosis.

### 6.5. Eye Absent Homolog 4 (EYA4)

The eye absent gene family consist of four members (*EYA1*, *EYA2*, *EYA3*, and *EYA4*) that are characterized by a highly conserved C-terminal EYA domain [181]. EYA proteins have phosphatase activity through their threonine and tyrosine residues on their different domains [53]. They are involved in different cellular functions, like cell proliferation and survival, cell migration, DNA damage repair, angiogenesis, and developmental cell polarity, among others [182]. In cancer, the EYA family has been related to the RTK/RAS/MAPK family and the TGF-β pathway, controlling cell apoptosis and proliferation [183]. Specifically, in CRC, *EYA4* has been suggested as a tumor suppressor involved in the inhibition of the WNT signaling pathway. *EYA4* is a possible candidate to be regulated by epigenetic mechanisms like DNA methylation. It has been proposed that the inactivation by aberrant promoter methylation affects its interaction with WNT, MAPK, and local adhesion pathways [184]. In esophageal squamous cell carcinoma (ESCC), its hypermethylated *EYA4* reduces the phosphorylation of AKT, therefore inducing EMT formation [185]. *EYA4* promoter hypermethylation may therefore serve as a biomarker for detection of CRC undergoing EMT.

### 6.6. GATA-Binding Protein 5 (GATA5)

GATA-binding proteins comprise a small family of zinc-finger transcriptions factors that bind to the GATA motif and are composed of six members ranging from *GATA1* to *GATA6* [186]. Through its highly conserved zinc-finger DNA-binding domain, it has been associated with various cell functions, such as gene regulation, epithelial differentiation, and organogenesis [187,188]. Moreover, studies have identified some GATA family members, specifically *GATA4*, *GATA5,* and *GATA6*, as tumor suppressor genes associated with several cancer diseases. In lung cancer, aberrant promoter DNA methylation leads to a loss of *GATA4* and *GATA5* activity as tumor suppressor genes [187]. In cholangiocarcinoma, deregulated *GATA5* gene expression has been correlated to cell proliferation and metastasis. This deregulation was thought to occur through DNA methylation changes, resulting in the inhibition of Wnt/β-catenin signaling activity [55]. DNA methylation changes of *GATA5* have also been observed in CRC and gastric cancer [189], marking it as a potential novel biomarker in early stages of colorectal cancer. It will be important to elucidate if it can also serve as a marker for metastasis in CRC, as was shown in other cancer types.

### 6.7. Brain-Derived Neurotrophic Factor (BDNF)

Brain-derived neurotrophic factor (*BDNF*) is a member of neurotrophin family, which includes Nerve Growth Factor (NGF), Neurotrophin 3 (NT-3), and Neurotrophin 4 (NT-4) [190]. All neurotrophins are secreted as precursors or proneurotrophin, which are cleaved in order to produce mature neurotrophin [191]. Neurotrophins like BDNF act as intracellular after binding to their receptors TrkB and p75NTR. This interaction activates the RAS signaling pathway and is associated to apoptosis resistance, high cell proliferation, and poor prognosis [192]. Activation of neurotrophins has also been associated with the promotion of cell survival and differentiation through PI3K/AKT activation [193]. Moreover, methylation status of *BDNF* has been linked with several adversities and diseases, like psychiatric vulnerabilities [194]. Furthermore, BDNF hypermethylation has been associated to poor prognosis and metastasis formation in CRC [88,151,152].

### 6.8. Transforming Growth Factor Beta Regulator 4 (TBRG4)

The FASTK protein family is involved in mitochondrial RNA regulator homeostasis [195]. It is comprised of six members named *FASTK* and *FASTKD1* to *FASTKD5* and characterized by three poorly conserved domains called FAST1, FAST2, and RAP [196]. The overexpression of FASTKD4/TBRG4 in different types of cancer is highly associated with cancer progression. In lung cancer, an increase in the expression of *TBRG4* is observed compared to normal tissue [197]. In esophageal cancer, high levels of *TBRG4* promote ESCC formation, by regulation of the levels of reactive oxygen species (ROS) and inhibition of the mitochondrial-dependent apoptotic signal [198]. Specifically in osteosarcoma, high levels of *TBRG4* corresponded to lower survival rates, promoting cell proliferation and invasion by upregulation of TGF-β and PI3K/AKT signaling [199]. New discoveries have found that *TBRG4* is hypomethylated in colorectal cancer, which highlight its capability to be a new putative biomarker in prognosis of metastasis [88]. However, there is not enough information yet, and further studies are needed to assess its potential as a metastasis biomarker.

### 6.9. Open Issues Regarding the Use of Methylation Biomarkers

There are many studies describing putative DNA methylation biomarkers for CRC metastasis. However, to date, none of them have made it to be used in clinical diagnosis. Some of the most promising candidates, including *INHBB*, *SMOC2*, *BDNF*, and *TBRG4,* have been tested in large patient cohorts and validated across groups and/or been identified by multiple studies. However, most of the candidate biomarkers described above currently lack large cohort studies and need to be clinically verified before they can be generally used. Some of the candidates, such as *BDNF* and *TBRG4*, are involved in important signaling pathways as PI3K/AKT and TGF-β. Studying their deregulation in more detail would not only benefit the development of novel biomarkers, but could also improve our understanding of the mechanisms involved in metastasis formation.

## 7. Conclusions

Cancer is a global health problem, affecting millions of people and saturating health systems worldwide. Although there is still no general cure for cancer, treatments are continuously improving and are significantly increasing the life expectancy of patients. Nevertheless, the aggressiveness of late-stage cancer and the difficulty of curing the disease highlights the importance for new strategies and methods to improve early discovery and therefore treatment success.

Novel DNA methylation biomarkers for the prognosis during the early stages of cancer, such as *INHBB*, *SMOC2*, *BDNF*, and *TBRG4*, are being tested for clinical use to improve detection methods and guide the treatment and diagnoses of cancer patients. Other interesting candidates, including *HLTF*, *GATA5*, and *EYA4*, still have to be tested in more detail. Furthermore, besides the clinical application, DNA methylation biomarkers may increase our understanding of the processes involved in metastasis formation.

Importantly, for common use of DNA methylation biomarkers in clinical diagnosis, detection methods need to be optimized and standardized. This will improve the sensitivity and specificity of the assays, which are currently still low. Together with the proposed biomarker validation described above, this will pave the way to establish methylation biomarkers in clinical diagnosis and prognosis.

## Figures and Tables

**Figure 1 biomolecules-11-01722-f001:**
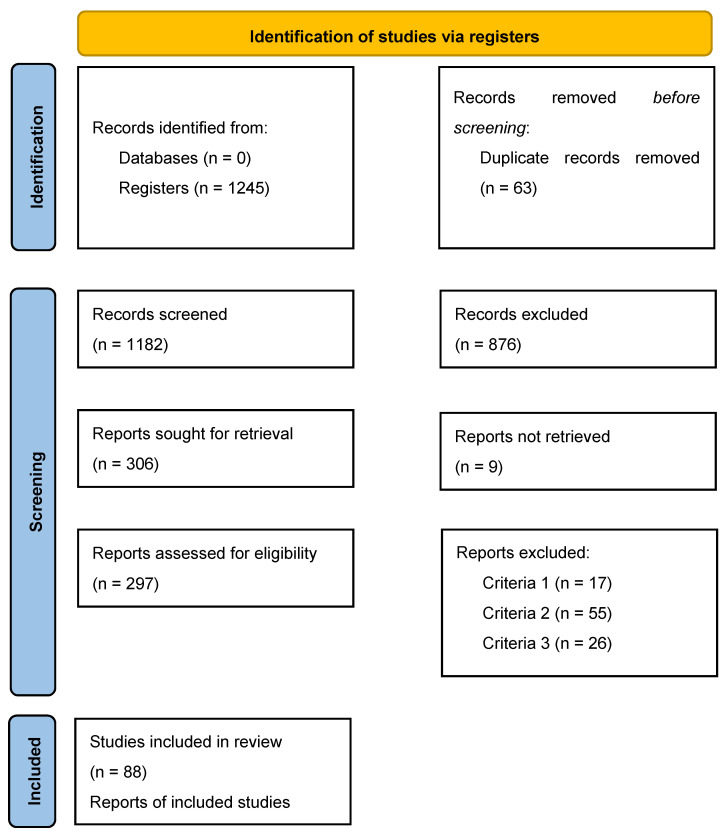
Flow diagram depicting the process of the identification, inclusion and exclusion criteria of selected studies.

**Table 1 biomolecules-11-01722-t001:** Methylation biomarkers in CRC diagnosis.

Gene	Pathways	Type of Marker	Samples	References
*ALX4*	Tumor suppressor of Wnt/β-catenin pathway [40]	Diagnosis	Plasma	[41,42,43,44,45,46,47]
*BCAT1*	mTOR1 regulator	Diagnosis/Prognosis	Plasma, FFPE	[41,44,48,49,50]
*BMP3*	Tumor suppressor via TAK1/JNK and ActRIIB/SMAD2-dependent gene [51]	Diagnosis	Plasma, Stool, FFPE	[41,43,46,48,49,52]
*EYA4*	Tumor suppressor of Wnt/β-catenin pathway [53]	Diagnosis/Prognosis	Plasma	[41,43,44,54]
*GATA4/GATA5*	Tumor suppressor of Wnt/β-catenin pathway [55]	Diagnosis	Plasma, Stool	[41,43,44,49,56]
*HIC*	Tumor suppressor of IL-6/STAT3 pathway [57]	Prognosis	Plasma	[41,43,44,56]
*HLTF*	Tumor suppressor TGF- β/SMAD pathway [58]	Diagnosis/Prognosis/Predictive	Plasma, Stool	[43,44,48,59]
*IGFBP-3*	Involved in TGF-β independent function [60]	Prognosis/Predictive	Plasma, FFPE	[42,45,54,61,62]
*IKZF1/IKZF2*	Related to binding to relevant genes as PTPN6 and MEIS2 [50,63]	Diagnosis/Prognosis	Plasma, FFPE	[41,44,48,49,50]
*IRF4*	Promotes proliferation through JNK/Jun pathway [64]	Diagnosis	Plasma	[41,44,49]
*ITGA4*	Increases migration through interacting with VCAM-1 [65]	Diagnosis	Stool, FFPE	[43,44,48,56]
*LINE-1*	Inserts into genes causing gene disruption [66]	Diagnosis/Prognosis	Plasma	[41,42,44,67]
*MGMT*	DNA repair system [68]	Diagnosis	Plasma, Stool, FFPE	[41,42,43,44,45,49,69]
*MLH1*	Member of mismatch repair (MMR) machinery [70]	Diagnosis	Stool, FFPE	[42,43,44,48,49,54,69,71,72,73]
*NDRG4*	Inhibition of PI3K/AKT pathway [74]	Diagnosis/Prognosis	Stool, FFPE	[41,43,48,49,52,56]
*OSMR*	Activation of STAT3/FAK/Src pathway [75]	Diagnosis	Plasma, Stool	[41,43,48,56]
*PRIMA1*	Activates mutant p53 pathway [76]	Diagnosis	Plasma	[41,49,52]
*RASSF1A/RASSF2A*	Interacts with Hippo and Wnt Pathway [77]	Diagnosis/Prognosis/Predictive	Plasma, Stool, FFPE	[41,43,44,49,52,78]
*SDC2*	Promotes MAPK and EMT pathway activation [79]	Diagnosis	Plasma, Stool	[41,44,45,46,49,52,56,80]
*SEPT9*	Activates Rho/ROCK and FAK pathway [81]	Diagnosis/Prognosis/Predictive	Plasma, FFPE	[41,42,45,46,48,49,52,54,82]
*SFRP1/SFRP2*	Inactivation of Wnt pathway [83]	Diagnosis	Plasma, Stool, FFPE	[41,43,44,45,48,49,52,56]
*TAC1*	-	Diagnosis/Prognosis/Predictive	Plasma	[41,44,54,82]
*TFPI2*	Involved in TWIST-integrin α5 [84]	Diagnosis	Plasma, Stool	[43,48,49]
*VIM*	Interacts with MAPK/ERK Pathway [85]	Diagnosis/Prognosis	Plasma, Stool, FFPE, Urine	[41,45,46,48,49,52,56]
*INHBB*	Activates ERK/Smad2/3 pathway [86]	Diagnosis/Prognosis	FFPE	[86]
*BDNF*	Activates AKT pathway [87]	Prognosis	FFPE	[88]
*SMOC2*	Activates Wnt/β-catenin pathway [89]	Prognosis	FFPE	[89]

FFPE: Formalin-Fixed Paraffin-Embedded tissue from Colorectal biopsy.

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
