# Peer review of "Novel Methylation Biomarkers for Colorectal Cancer Prognosis"

_biomolecules, 2021, doi:10.3390/biom11111722_

Round 1

Reviewer 1 Report

This manuscript summarizes the latest findings on epigenetic changes (mainly methylation) of genes involved in colorectal cancer metastasis.

Broad comments:

Although this manuscript seems to describe well the information of novel genes related to metastasis of colorectal cancer and involved in methylation, it is somewhat problematic to consider them as biomarkers.

  1. The authors pointed to the lack of biomarkers for metastasis as a problem. I agree with this point, but the main reason for this is the lack of detection methods. The authors also point this out (page 2, line 54), but there is a lack of discussion on this point. In reality, we can only measure the concentration of products leaked into the blood due to overexpression or underexpression caused by methylation changes. For example, an overview of methods to detect methylation and new (possible) methods to detect it would be helpful to motivate the development of new detection methods in the future.

  1. An overview of biomarkers is given in the Introduction (lines 41-55), but from the perspective of the particular event of metastasis, I have the impression that the description contains contradictions. In my view, biomarkers in general are good at capturing the presence or absence of disease occurrence, but not good at tracking the stages of the disease. Metastasis is a relatively late event, which makes the adaptation of biomarkers even more difficult. I would like to see an overview of biomarkers for the specific event of metastasis, rather than a general overview of them.

Specific comments:

  1. Abstract, line 10; since the pathways involved are not limited to metabolism, "signaling pathways" might be a more appropriate term.
  2. Line 68; PI3KCA→PIK3CA?
  3. Line 118; isn't "almost exclusively" an overemphasis? For example, since lung metastasis of prostate cancer is seen even in a small number of cases, it would be better to say "mostly".
  4. Line 203; the phenotypes such as CIN, MSI, and CIMP are not classified by differences in epigenetic changes.

Reviewer 2 Report

The authors describe this review paper on the topic of novel methylation biomarkers in colorectal cancer [rognosis . Overall, the description is powerful and well written. The review article give a solution  of such biomarkers  in metastasis of CRC that may allow better treatment and early identification of cancer formation in order to perform better diagnostics . For this reason, the paper deserves such a credit for carrying out such a study. I agree the conclusions of methylation biomarkers may be regarded as the potential treatments to CRC in the future.              

Reviewer 3 Report

The aim of this narrative review is to describe their current understanding of the mechanisms leading to colorectal cancer and metastasis formation and discuss the role of DNA methylation in these pathways, and the increasing knowledge in these processes has led to the identification of biomarkers for the prognosis of CRC and metastasis. Furthermore, they also present and discuss novel biomarkers that have the potential to change our treatment strategies for CRC in the future. These contents seems informative and appealing; however, there are a lot of criticisms and have several issues that the authors need to address before the manuscript is suitable for publication.

Major Compulsory Revisions:     

  1. A flow diagram depicting the process of the identification and inclusion of selected studies for this narrative review is mandatory. And the corresponding paragraph of Methods to the flow diagram needs be added in the text.
  2. As the manuscript title is Novel Methylation Biomarkers for Colorectal Cancer Prognosis, authors were suggested to focus on the relevant information and discussion and the redundant paragraphs should be condensed or omitted, for example, in 1. Introduction section: Although biomarkers have become robust and efficient molecules to evaluate potential abnormal biological process, the detection methods, such as polymerase chain reaction (PCR), enzyme linked immunosorbent assay (ELISA), surface plasmon resonance (SPR), still lack some accuracy, sensitivity and specificity in clinical diagnosis, the methodology for the detection of DNA methylation is lack; Insights into Colorectal Carcinogenesis section: paragraph 1; 3. From CRC to Metastasis: A complex journey: paragraphs 1-2 and 5. An extensively revised was mandatory to fulfill their review objectives. I strongly authors cite the references and reformat the overall structure following these article: A DNA methylation based biomarkers in colorectal cancer: A systematic review. Biochim Biophys Acta 2016;1866:106-20; Hypermethylated DNA as a biomarker for colorectal cancer: a systematic review. Colorectal Dis 2016;18(6):549-61.
  3. Table 1. Methylation biomarkers in CRC detection. In this table, not only biomarkers for CRC detection, diagnosis, prognosis and prediction were also suggested to be included. Predictive biomarkers were not discussed, e.g. Methylation of IGFBP3, mir148a and PTEN are found to be predictive markers for 5-FU and EGFR therapy respectively (A DNA methylation based biomarkers in colorectal cancer: A systematic review. Biochim Biophys Acta 2016;1866:106-20). The difference between detection and diagnosis biomarkers in CRC? Please uniform the wording in column of Type of marker to be noun or adjective. Please search the database in more details as a lot of relevant references were missing, e.g. MLH1 and MGMT, Correlation of MLH1 and MGMT methylation levels between peripheral blood leukocytes and colorectal tissue DNA samples in colorectal cancer patients. Oncol Lett 2013;6(5):1370-1376; LINE-1, Methylation Status Correlates Significantly to Post-Therapeutic Recurrence in Stage III Colon Cancer Patients Receiving FOLFOX-4 Adjuvant Chemotherapy. PLoS One. 2015 Apr 28;10(4):e0123973; SDC2, Stool DNA test targeting methylated syndecan-2 (SDC2) as a noninvasive screening method for colorectal cancer. Biosci Rep 2021;41(1):BSR20201930; ALX4, BMP3, NPTX2, RARB, SDC2, SEPT9, and VIM, Hypermethylated DNA, a circulating biomarker for colorectal cancer detection. PLoS One 2017;12(7):e0180809, etc.
  4. Novel Methylation Biomarkers in CRC and Metastasis Prognosis section: How authors define the term of novel methylation biomarkers in CRC and metastasis?
  5. An overall Discussion section is mandatory for some unresolved issues, such as, individual hypermethylated DNA promoter regions have limited value as CRC screening markers. However, a panel of seven hypermethylated promoter regions show great promise as a model for CRC detection (Hypermethylated DNA, a circulating biomarker for colorectal cancer detection. PLoS One 2017;12(7):e0180809), and I suggest authors to add this view point into the Discussion section.
  6. The Conclusion section did not completely reflect the main text part and should be amended and augmented.
  7. An important goal of this review article is to present and discuss novel biomarkers that have the potential to change our treatment strategies for CRC in the future; however, the relevant information was rarely discussed in more details.

Round 2

Reviewer 3 Report

Authors have replied all queries adequately and could be considered as acceptable for publication.